# Toward Precision Weight-Loss Dietary Interventions: Findings from the POUNDS Lost Trial

**DOI:** 10.3390/nu15163665

**Published:** 2023-08-21

**Authors:** Lu Qi, Yoriko Heianza, Xiang Li, Frank M. Sacks, George A. Bray

**Affiliations:** 1Department of Epidemiology, School of Public Health and Tropical Medicine, Tulane University, New Orleans, LA 70118, USA; 2Department of Nutrition, Harvard T.H. Chan School of Public Health, Boston, MA 02115, USA; 3Department of Clinical Obesity, Pennington Biomedical Research Center, Louisiana State University, Baton Rouge, LA 70808, USA

**Keywords:** BMI, body mass index, individual variability, dietary interventions, precision health

## Abstract

The POUNDS Lost trial is a 2-year clinical trial testing the effects of dietary interventions on weight loss. This study included 811 adults with overweight or obesity who were randomized to one of four diets that contained either 15% or 25% protein and 20% or 40% fat in a 2 × 2 factorial design. By 2 years, participants on average lost from 2.9 to 3.6 kg in body weight in the four intervention arms, while no significant difference was observed across the intervention arms. In POUNDS Lost, we performed a series of ancillary studies to detect intrinsic factors particular to genomic, epigenomic, and metabolomic markers that may modulate changes in weight and other cardiometabolic traits in response to the weight-loss dietary interventions. Genomic variants identified from genome-wide association studies (GWASs) on obesity, type 2 diabetes, glucose and lipid metabolisms, gut microbiome, and dietary intakes have been found to interact with dietary macronutrients (fat, protein, and carbohydrates) in relation to weight loss and changes of body composition and cardiometabolic traits. In addition, we recently investigated epigenomic modifications, particularly blood DNA methylation and circulating microRNAs (miRNAs). We reported DNA methylation levels at *NFATC2IP*, *CPT1A*, *TXNIP*, and *LINC00319* were related to weight loss or changes of glucose, lipids, and blood pressure; we also reported thrifty miRNA expression as a significant epigenomic marker related to changes in insulin sensitivity and adiposity. Our studies have also highlighted the importance of temporal changes in novel metabolomic signatures for gut microbiota, bile acids, and amino acids as predictors for achievement of successful weight loss outcomes. Moreover, our studies indicate that biochemical, behavioral, and psychosocial factors such as physical activity, sleep disturbance, and appetite may also modulate metabolic changes during dietary interventions. This review summarized our major findings in the POUNDS Lost trial, which provided preliminary evidence supporting the development of precision diet interventions for obesity management.

## 1. Introduction

Obesity emerged during the late 1970s and has become epidemic in the United States and Westernized countries, with the rapid escalation of a variety of complicating chronic diseases such as type 2 diabetes, cardiovascular disease, and cancer, as well as premature mortality [1,2]. Obesity management plays a pivotal role in mitigating the risks of complicated disorders [3]; weight-loss dietary intervention has become a fundamental strategy for obesity management [4]. Traditional dietary interventions are ‘one dose fits all’ methods, assuming that individuals respond to the interventions following the same manner. However, findings from human genomic research and investigations on gene–diet interactions have revealed that considerable individual variability exists in the relations between dietary intakes and body weight changes; such variability is substantially determined by genetic variations [5,6,7]. In addition, growing studies have also implicated other intrinsic factors including epigenomic modifications, metabolomic profiles, and gut microbiome compositions in determining the inter-personal heterogeneous responses to diet.

In 2015, the Precision Medicine Initiative was launched, with a target to develop novel prevention and treatment methods that may have upgraded efficacy by taking individual variability into account [8]. The principle of precision medicine has been applied in various research areas, including nutrition, initiating precision nutrition [9]. Precision nutrition aims to provide evidence for developing precision dietary interventions, which may improve the effects in disease management and prevention [9]. Studies on gene–diet interactions gleaned from observational studies contribute greatly to understanding the potential role of genetic variants in determining individual variability [6,7,10,11,12]. However, the inherent limitations of observational studies may introduce noise into the findings due to various biases [13]. Randomized clinical trial has been widely accepted as the gold standard for causal inference [14], And is therefore an ideal setting for assessing the potential roles of various intrinsic factors in determining an individual’s response to dietary intakes [15]. The roles of genomics and other intrinsic factors in modulating the dietary effects on weight loss and cardiometabolic outcomes have been increasingly studied in existing dietary intervention trials [16,17,18]. The Preventing Overweight Using Novel Dietary Strategies (POUNDS Lost) trial, one of the largest and longest weight-loss intervention trials, tested four diets varying in macronutrient intakes [19]. In POUNDS Lost, considerable inter-personal variability was observed in weight changes in response to weight-loss dietary interventions, similar to other clinical trials for weight loss [20,21]. By leveraging the rich sources of clinical and omics measurements in POUNDS Lost, we have performed a series of studies to investigate various omics variations in relation to diet intervention-induced weight loss and metabolic changes among adults with overweight or obesity. 

This review summarized the landscape and major findings from the milestone studies that have been performed in the POUNDS Lost trial, with a focus on testing the genomic, epigenomic, and metabolomic factors and their interactions with dietary interventions in relation to weight loss and changes of a variety of cardiometabolic outcomes. The review also included studies addressing the roles of biochemical, behavioral, and psychological factors in modulating responses to diet interventions, beyond the omics factors. 

## 2. The POUNDS Lost Trial

The POUNDS Lost trial included 811 adult men and women with overweight or obesity (BMI, 25–40 kg/m^2^). Participants were randomly assigned to one of four diets varying in macronutrient compositions, which constitutes a 2 × 2 factorial design (Table 1). Such design also generated a graded difference in carbohydrates from 35% to 65% in the diets tested. Calorie intake was restricted by 700 kcal/day below energy requirements, based on measuring the resting energy expenditure using a metabolic cart and multiplying it by an activity factor. All four diets were low in saturated fat and were in accordance with cardiovascular health guidelines. At both sites, a behavioral program with consistent content and intensity was implemented uniformly across the four groups. 

In POUNDS Lost, multi-dimensional approaches were applied to assess behavioral and dietary adherence [22]. A computer tracking system collected data on eight indicators of adherence throughout the first 6 months. Compliance with the diet was assessed by 24-h dietary recalls conducted via telephone interviews at 6 and 24 months in a random 50% subset of participants at both sites. Individuals who achieved their assigned goals within a range of ± 5% were labeled as “adherent.” A total of 241 participants wore pedometers at their belt line, aiming to promote adherence to the exercise program. Number of steps were recorded from this subset across each of the 4 dietary groups at baseline and 6 months. The prescription of a high-fat diet (vs. a low-fat diet) was found to be associated with higher levels of dietary adherence, though behavioral adherence was similar across the four intervention groups [22].

In POUNDS Lost, participants had lost an average of 6 kg in body weight at 6 months, but began to regain weight after 12 months [19]. Such a weight change trajectory has also been observed in other long-term weight-loss trials. Among participants who completed the trial (80% of the study population), the average weight loss was 4 kg. Analysis of the imputed data revealed consistent mean weight losses. Weight loss was 3.0 kg in the average-protein group, 3.6 kg in the high-protein group, and 3.3 kg in both diet groups assigned to the low- and high-fat diets at 2 years, respectively. The weight losses with the highest and lowest carbohydrate were 2.9 vs. 3.4 kg over 2 years. Similarly, no difference was observed in weight loss or weight maintenance at 2 years in the intent to treat analysis. Significant variation in weight loss among participants was found within each dietary intervention group [21].

## 3. Gene–Diet Interactions

The individual variability in the development of obesity and metabolic responses to diets are fundamentally affected by genomic variants [10,23]. To date, genome-wide association studies (GWASs) have identified nearly a thousand genetic loci harboring the variants related to the common form of obesity and body fat distribution [5,24,25]. In addition, growing GWASs have also identified variants associated with various obesity-related diseases such as type 2 diabetes, cardiovascular disease, and cancer, [26] as well as dietary factors [27,28,29]. In the first GWASs on intakes of dietary macronutrients, we found the genetic variant in the *FGF21* gene was associated with caloric intakes from carbohydrates and fat, and the variant in the *FTO* gene was associated with caloric intake from protein [27]. The following GWAS also linked genetic variants to intakes of various nutrients, foods, and beverages [29,30]. Notably, the effect size associated with individual genetic variant is typically very moderate. In addition to the main associations, a wealth of data from observational studies have revealed that genetic variants may interact with dietary factors in relation to adiposity, weight change, and risk of obesity [6,7,11]. However, the findings from observational studies may be biased by potential confounding effects of other diet-correlated factors, such as lifestyle and socioeconomic factors, reverse causation, and inaccuracy in nutrition assessment. In contrast, clinical trials testing the effects of dietary interventions typically assess prescribed diets on the longitudinal changes of the outcomes, and therefore, may minimize these biases. Indeed, evidence-based prevention and treatment rely mainly on the statistical interpretation of data from properly conducted clinical trials, which occupy the top tier of evidence in most scales [31].

In 2011, we published the first study testing gene–diet interactions on weight loss in the POUNDS Lost trial [32]. In this study, we reported that high-carbohydrate and low-fat diet intervention was related to better weight loss and improvement of insulin sensitivity in individuals with the rs2943641 CC genotype in the *IRS1* gene than those without this genotype. To date, we have published nearly 50 papers with a focus on gene–diet interactions on weight loss and changes in a variety of cardiometabolic outcomes including fat distribution, blood pressure, glucose, insulin resistance, appetite, and measures of energy expenditure in POUNDS Lost [21]. The genetic variants tested include those near the genes related to obesity (*FTO*, *MC4R*) [33,34,35], diabetes (*TCF7L2, HNF1A*) [36,37] and metabolisms of glucose (*IRS1*, *GIPR*) [32,38], lipids (*APOA5*, *LIPC*, *CETP*) [39,40,41], and amino acids (*PPM1K*) [42], as well as the genes related to gut microbiota (*LCT*) [43], circadian rhythm (*MTNR1B, CRY2*) [44,45], and dietary intakes of energy and macronutrients (*AMY1-AMY2*, *FGF21*) [46,47]. 

It has been noted that combining multiple variants together is associated with greater variance in phenotypes, motivating the analysis on the joint genetic effects by creating genetic risk scores (GRSs) that combines numerous genetic variants of interest and reflect the overall genetic susceptibility [48]. In a study, we created a GRS based on three genetic variants near the *DHCR7*, *CYP2R1*, and *GC* genes that have been related to circulating vitamin D levels. Such vitamin D GRSs showed significant interaction with dietary fat intakes on 2-year changes in whole-body bone mineral density (BMD). Higher GRS was associated with significantly less reduction in whole-body BMD than those with lower GRSs in the high-fat diet group, whereas no genetic associations were observed in the low-fat diet group [49]. In another study, we analyzed a GRS for diabetes, and found it significantly interacted with dietary protein in relation to biomarkers of glucose metabolism, the homeostasis model assessment of insulin resistance (HOMA-IR), and β cell function (HOMA-B) at 2 years in white Americans [50]. Gut microbiota plays a key role in regulating adiposity. In a recent study, we created a GRS for gut microbial composition [51]. We found that higher GRS for microbial abundance was significantly associated with greater reductions in waist circumference, total fat mass (FM), whole-body total percentage of fat mass (FM%), and percentage of trunk fat (TF%) at 2 years. The microbial GRS also showed significant interactions with dietary protein in relation to changes in total FM, FM%, and TF% at 6 months [52]. We also reported GRSs for blood pressure, [53] lean body mass, [54] adiposity subtypes, [55] and short chain fatty acids (SCFAs) [56] with changes in glycemic markers, appetite, blood pressure, and bone mineral density in response to diet interventions. 

A critical challenge in studying gene–diet interaction in clinical trials is the lack of replications and low study power. To overcome these shortcomings, we have made efforts to collaborate with other diet and lifestyle intervention trials. In collaboration with a 2-year weight-loss trial—Dietary Intervention Randomized Controlled Trial (DIRECT), we have replicated the findings of interactions between the CETP variant and dietary fat on changes of HDL cholesterol and triglycerides during diet interventions in POUNDS Lost [41]. In these two trials, we also observed consistent interactions between the HNF1A variant and dietary fat intakes on weight loss and reduction in waist circumference [37]. In a larger scale collaboration including eight randomized controlled trials (*n* = 9563), we tested associations of the FTO genotypes with changes in various adiposity measures during interventions. No significant associations were observed. We did not find interactions of the FTO genotypes with intervention type and length in relation to the outcomes [16].

## 4. Epigenetic Modifications—DNA Methylation

Epigenomic modifications refer to alternations in gene functionality that do not involve any modification in the DNA sequence, including DNA methylation, histone modification, and non-coding RNA regulation [57]. Because epigenetic changes are modifiable in nature, it has piqued our interest as potential targets for interventions. The field of epigenetics has been primarily focused on studying DNA methylation. DNA methylation, a heritable and reversible epigenetic modification, entails appending of a methyl group to the DNA structure. DNA methylation acts in the interface of gene and environmental stressors, therefore influencing the subsequent gene expressions [58]. Epigenome-wide association studies (EWASs) have identified genes with DNA methylation levels related to body mass index and obesity [59,60]. Growing evidence has shown that DNA methylations are subject to modulation by dietary factors [58,61,62]. In POUNDS Lost, we have performed a series of studies investigating the relationship between pre-treatment regional DNA methylations around different genes with various cardiometabolic changes (glycemic traits, lipids, and blood pressure) in response to dietary weight loss interventions with different macronutrient compositions [63,64,65,66]. 

The protein coding gene *TXNIP* plays a critical role in glucose metabolism and regulation of β cell function (Figure 1) [67,68]. Methylation of cg19693031 at *TXNIP* is the CpG site most robustly associated with type 2 diabetes in previous epigenome-wide association studies [69,70]. Data from the POUNDS lost trial confirmed that higher levels of DNA methylation at *TXNIP* were correlated with lower levels of glucose, HbA1c, and HOMA-IR cross-sectionally at baseline [63]. Interestingly, for participants who were in the top tertile of DNA methylation at *TXNIP*, an average-protein (15%) diet was associated with more substantial decreases in insulin and HOMA-IR levels, whereas no differences were observed in the lower tertiles (Figure 1) [63]. 

Elevated concentrations of triglycerides and triglyceride-rich lipoproteins, such as very low-density lipoprotein (VLDL), are commonly observed in obesity and are associated with a heightened risk of developing cardiometabolic abnormalities. Evidence from epigenome-wide association studies has consistently reported methylation of several CpG sites at *CPT1A* with lipid traits and body mass index [71]. In the POUNDs Lost trial, the regional DNA methylation at *CPT1A* exhibited different associations with 2-year changes in total plasma triglycerides among participants assigned to low- or high-fat weight loss diets. In the low-fat group, but not the high-fat diet group, higher DNA methylation at the region around 3 CpG sites at *CPT1A* was associated with greater reductions in total plasma triglycerides at 2 years, independent of concurrent weight loss [64]. Considering VLDL is rich in triglycerides, further investigations were conducted for lipids and lipoproteins in VLDL, which revealed similar interactions for 2-year changes in VLDL-triglycerides, VLDL-cholesterol, and VLDL-apolipoprotein B, but not for VLDL-apolipoprotein C-III. 

*LINC00319* (cg01820192) is one of the few CpG sites that have been linked to both birthweight and hypertension [72,73]. Evidence has shown that prenatal adversity exposures could affect DNA methylation to facilitate metabolic memory and trigger future metabolic abnormalities in adulthood [74,75]. In POUNDs lost, we analyzed DNA methylation at several loci *LINC00319*, *ATP2B1*, and *LMNA*, which have been related to both birth weight as an indicator of prenatal adverse exposures, and blood pressure in prior EWASs [72,73]. A significant interaction was found between dietary fat intake and regional DNA methylation at *LINC00319* on 2-year changes in blood pressure [65]. A higher level of DNA methylation was associated with a greater reduction in both systolic (SBP) and diastolic (DBP) blood pressures among those assigned to a low-fat diet, but not for those assigned to a high-fat diet. Moreover, participants with higher DNA methylation were more likely to exhibit declining trends in SBP and DBP. These findings from the POUNDS lost trial suggest that epigenetic modifications play an important role in regulating metabolic changes in response to dietary weight-loss interventions. While the exact mechanisms are still being studied, these findings suggest individual pre-treatment DNA methylation profiles may partly explain the large inter-individual variabilities in response to intervention and could potentially be targeted in the development of new weight loss interventions.

## 5. Epigenetic Modifications—Thrifty microRNA 

In 1962, James Neel proposed thrifty genotype hypothesis, which suggested that exposure to periods of famine during human evolutionary history resulted in selection pressures in favor of a thrifty genotype that led to highly efficient fat storage during periods of abundance [76]. Evidence exists supporting the genotype of the *LCT* gene, which encodes enzyme lactase, as thrifty genotype [77]. The *LCT* is a hallmark gene of positive selection, and the *LCT* genotype determines the ability to digest fresh milk, a rich source of protein and fat, to survive famine, by promoting energy storage and adiposity. In the POUNDS Lost trial, we performed studies on the thrifty *LCT* genotype and epigenetic modifications (Figure 2) [43,78]. We observed that in the POUNDS Lost, a participant having a higher number of the G allele of *LCT*-rs4988235 was correlated with a lower degree of general adiposity, and the G allele was related to lower milk intake among participants with data on food records at the baseline examination [43]. Changes in adiposity and energy expenditure in response to weight-loss dietary intervention are closely correlated in the participants in this trial [79,80], while it remains unclear whether such a thrifty genotype affect changes in adiposity and energy expenditure. 

In a recent study, miR-128-1 was identified as a potential thrifty miRNA located on chromosome 2q21.3 near the *LCT* gene [81]. We recently completed circulating miRNA measurements in POUNDS Lost and investigated the function and implications of a newly described circulating miR-128-1 in relation to weight loss and cardiometabolic changes [78]. miRNAs are short non-coding RNAs that post-transcriptionally regulate gene expression. Circulating miRNAs in serum/plasma are emerging novel biomarkers of obesity, metabolic diseases, and cardiovascular health [82,83,84], by coordinating the regulation of whole-body metabolism through intercellular communications. We found that circulating thrifty miRNA was associated with achieving successful weight-loss outcomes and might become a novel target of diet and lifestyle interventions to improve adiposity, insulin resistance, and energy metabolism. These data highlight the need for considering individuals’ genetic and epigenetic adaptation to nutrient digestion to improve the efficacy of dietary interventions. 

## 6. Metabolomics, Gut Microbiota Metabolites, Bile Acids, and Amino Acids

Identifying novel biological pathways associated with improved metabolic conditions in patients with overweight and obesity is a critical research concern for developing therapeutic targets and preventive approaches. The ongoing targeted and untargeted metabolomics analysis projects have provided significant associations of weight-loss diet-induced changes in metabolites with long-term successful weight-loss outcomes, including cardiometabolic risk factors (such as atherogenic lipoprotein metabolism, adiposity, insulin resistance, and glucose metabolism) by repeatedly measuring circulating metabolites (Figure 3). 

Dietary intakes and different macronutrients (such as fat and protein) are among the most important regulators of gut microbiota, [85] and circulating gut-microbiota-related metabolites (such as trimethylamine N-oxide (TMAO) and its precursors), amino acids, and bile acids changed dynamically in response to weight-loss diets among the POUNDS Lost study participants. Recent studies have identified that atherogenic metabolites are generated by the microbial metabolism of foods rich in dietary protein and fat, amino acids, and other nutrient precursors [86,87]. In the POUNDS Lost trial, 6-month changes in atherogenic gut-microbial metabolites, TMAO or its precursor metabolites L-carnitine and choline, are associated with long-term weight loss, and improved glucose metabolism, insulin sensitivity, and atherogenic lipoproteins in response to weight-loss diet interventions. Changes in circulating nutrient precursors of TMAO were significantly predictive of successful long-term weight loss, suggesting that these gut-microbial metabolite changes could be biomarkers for assessing the effectiveness of a long-term dietary intervention. Moreover, weight-loss diet-induced changes in TMAO and its precursor metabolites (choline and L-carnitine) were significantly related to the improvement in insulin sensitivity. When we examined potential interaction associations between TMAO changes and macronutrient intakes, results showed that habitual intake of dietary fat modified the associations of TMAO changes with changes in glucose and insulin sensitivity [88]. 

Branched-chain amino acids and aromatic amino acids (AAAs) have been related to the gut microbiome and may be linked to the development of obesity and insulin resistance. Findings from the POUNDS Lost suggest the potential importance of dietary interventions in modifying amino acid profiles along with and beyond weight loss [89]. Analysis of ectopic fat storage and fat distribution measured by repeated CT scans, weight-loss diet-induced decreases in valine, leucine, and isoleucine, and total BCAAs were associated with reductions in hepatic and abdominal fat during the 2-year intervention course [90]. We comprehensively examined how alterations in diabetes-related amino acids might influence associations of changes in gut microbial metabolites with improvements in glycemia and insulin sensitivity in weight-loss dietary intervention. In the POUNDS Lost trial, the associations of changes in TMAO, choline, and L-carnitine with diabetes-related traits were independent of changes in the amino acids. These suggested that the observed results are possibly through different pathways or mechanisms related to diabetes-related amino acids [91], and the independent roles of the TMAO-related metabolism in regulating the metabolic responses to weight-loss diet interventions. 

Bile acids (BAs) have been recently recognized as endocrine signaling molecules that regulate lipid homeostasis and other metabolic conditions related to obesity by activating BA-responsive nuclear receptors [92,93]. Primary BAs are synthesized from cholesterol in the liver and conjugated to the amino acids glycine or taurine before secretion. Gut microbiota metabolizes primary BAs to generate secondary Bas [94,95]. We have performed the metabolomics platform C18-neg to measure circulating plasma BA subtypes using a liquid chromatography-tandem mass spectrometry, and investigated whether weight-loss diet-induced changes in circulating BA subtypes were associated with the improvements in adiposity, body composition, fat distribution, energy metabolism, and cardiometabolic risk factors, such as glucose metabolism, insulin sensitivity, lipids, and Atherosclerotic Cardiovascular Disease (ASCVD) risk [96]. 

Our study showed that decreases in the primary bile acids, which are synthesized in the liver, and the secondary bile acids produced by the gut microbiota were significantly associated with changes in body adiposity and energy metabolism in response to the interventions. Further, our study showed that early changes in specific primary and secondary bile acid subtypes during the first 6 months of intervention significantly predicted successful weight loss (5% or more weight reduction) over 2 years [97]. Decreases in specific primary and secondary bile acid subtypes in response to interventions were related to improved hyperinsulinemia and insulin resistance [96]. Our ongoing metabolome-wide projects aim to detect more novel circulating microbial metabolites associated with long-term weight loss in response to diet interventions. Also, different macronutrients, such as fat, carbohydrate, and protein intakes, may modify such associations, implicating the importance of precision nutrition strategies to improve cardiometabolic conditions in response to weight-loss diet interventions for patients with obesity. 

## 7. Biochemical, Behavioral and Psychological Factors 

A group of biochemical factors relevant to body weight and related cardiometabolic traits have been measured in POUNDS Lost. Adiponectin is an adipokine protein secreted by adipose tissue that plays a pivotal role in regulating whole-body metabolism. Elevation of circulating adiponectin has shown effectiveness in preventing obesity and related cardiometabolic disorders; therefore, adiponectin is considered a therapeutic target for obesity, diabetes, and endothelial dysfunction [98]. In POUNDS Lost, weight-loss dietary interventions significantly increased circulating adiponectin levels over 2 years similarly in the four diet groups. We found that the increase of adiponectin was associated with reduction of waist circumference and LDL cholesterol, but with an increase of HDL cholesterol, after adjusting for concurrent weight change and other covariates [99]. In another study, we measured thyroid hormones (free triiodothyronine (T3), free thyroxine (T4), total T3, total T4 and thyroid-stimulating hormone (TSH)) in blood. Higher baseline free T3 and free T4 levels were significantly associated with a greater weight loss during the first 6 months [100]. 

Physical activity (PA) plays a pivotal role in regulating adiposity. In POUNDS Lost, PA was measured objectively with pedometers. We found that increment of PA was associated with greater reduction in body weight, waist, body composition, and fat distribution from baseline to 6 months [101]. The patterns of PA change were also associated with the trajectory of adiposity changes. Participants with the largest increase in PA maintained weight loss, while those with a smaller increase in PA regained weight, from 6 months to 24 months. We also found that the associations between PA and adiposity changes over 24 months were modified by dietary fat or protein intake. In addition, growing evidence has consistently implicated sleep behaviors in regulating adiposity. Recently, we analyzed the relation of changes in sleep disturbance with changes in body weight [102]. Compared with individuals without sleep disturbance, participants with slight, moderate, or great sleep disturbance showed an elevated probability of successful weight loss failure at 6 months, respectively. 

In another study, Liu et al. examined the associations between psychological and behavioral predictors with weight changes and energy intake [103]. It was found that every 1-point increase in craving score for high-fat foods at baseline was associated with greater weight loss and a decrease in energy intake and fat intake during the weight-loss period. In contrast, craving for carbohydrates/starches was associated with both less weight loss and more weight regain. In addition, it was found that greater cognitive restraint of eating was associated with less weight loss and more weight regain, whereas greater disinhibition of eating was only associated with more weight regain. 

## 8. Summary and Future Directions 

In the POUNDS Lost trial, we have comprehensively assessed the role of genomic variations, epigenomic modifications (DNA methylation and miRNAs), metabolomic analytes, and biochemical, behavioral, and lifestyle factors in relation to weight loss and cardiometabolic changes in response to dietary interventions (Figure 4). To date, the majority of our studies in the POUNDS Lost trial primarily address the role of various omics markers separately in modulating inter-personal variability in changes of body weight and related metabolic outcomes during dietary interventions. In fact, the effects of various omics mechanisms may not be separate, but act in a concerted manner [104]. For example, the genomic variants may modulate metabolomic and proteomic profiles, and epigenomic modifications may link the gut microbiome with the health outcomes such as obesity and diabetes [105]. Therefore, the analysis of any single omics may not provide complete picture of the underlying biology [106], and the integrated investigation to connect the multi-tiered omics profiles has become a trend for a deeper understanding of how nutrition affects precision health [107]. The multi-omics approaches have been recently applied to identify the subtypes of patients [108]. Integration of genome with the gut microbiome, epigenome, metabolome, and proteome can help identify the subgroups of individuals responding differently to the defined diet interventions [9]. Such systematic classification holds advantages over the individual omics markers to account for a larger proportion of individual variability in response to dietary interventions. Further research is warranted to address how the distinct systems-classified subtypes respond to diets. Compelling evidence has indicated that a variety of weight-loss dietary interventions generally benefit the whole population, while the precision dietary interventions that take the individual variability into account hold great promise to improve the efficacy and maximize the dietary effects in preventing obesity and mitigating its adverse complications.

Different from observational studies, the unique intervention strategies used in various clinical trials make it challenging to find identical trials for validating findings from dietary intervention trials. Even so, investigating the intrinsic determinants in multiple trials merits further efforts. First, previous collaborations between POUNDS Lost and other intervention studies have demonstrated improved study power. Second, the complementary measures of body adiposity and metabolic traits benefit indirect validations. Collaboration including multiple clinical trials with sufficient power will provide an ideal platform, particularly for multi-omics analyses and would be a productive next step. The findings from such investigations will be the basis for the development of novel, precision interventions for obesity management that are better tailored to meet the individuals’ needs in the future. 

## Figures and Tables

**Figure 1 nutrients-15-03665-f001:**
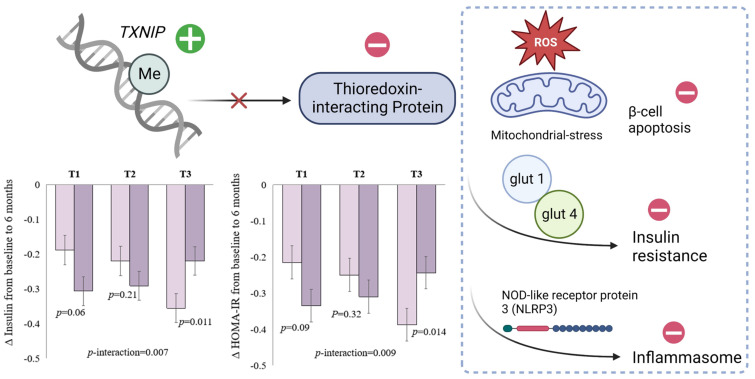
DNA methylation at *TXNIP* and schematic function of thioredoxin-interacting protein. Figure adapted with permission from Ref. [63] Copyright © 2022, Li X et al, under exclusive licence to Springer Nature Limited.

**Figure 2 nutrients-15-03665-f002:**
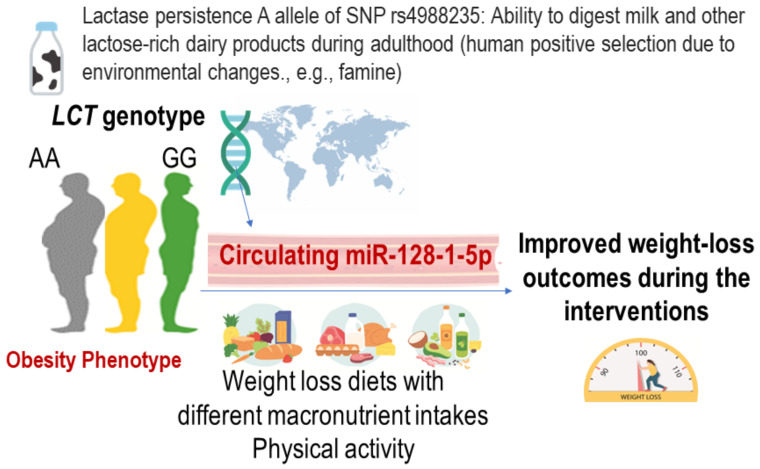
Possible roles of *LCT* genotype and circulating thrifty miRNA 128-1-5p in relation to the achievement of improved metabolic outcomes.

**Figure 3 nutrients-15-03665-f003:**
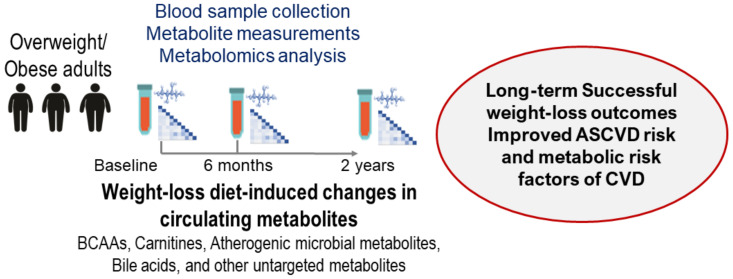
Overview of the metabolomics analysis projects in the POUNDS Lost trial.

**Figure 4 nutrients-15-03665-f004:**
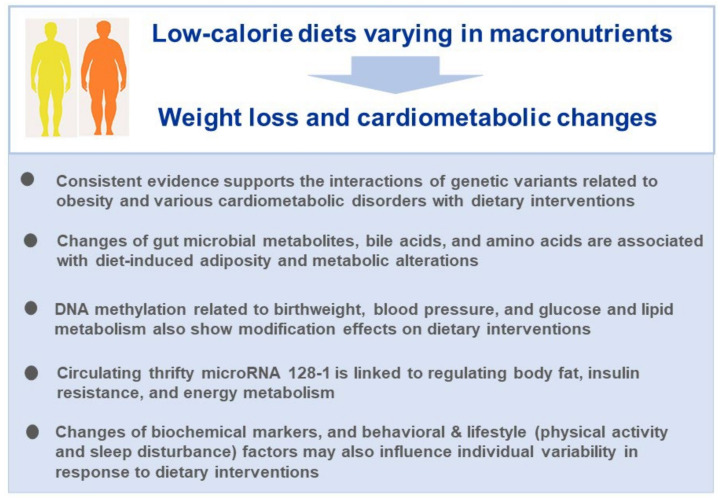
Take home messages from the studies in the POUNDS Lost trial.

**Table 1 nutrients-15-03665-t001:** Weight-loss diets tested in the POUNDS Lost trial.

Diets	Fat (%)	Proteins (%)	Carbohydrates (%)
low-fat, average protein	20	15	65
low-fat, high protein	20	25	55
high-fat, average protein	40	15	45
high-fat, high protein	40	25	35

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
