# Peer review of "Toward Precision Weight-Loss Dietary Interventions: Findings from the POUNDS Lost Trial"

_nutrients, 2023, doi:10.3390/nu15163665_

Round 1

Reviewer 1 Report

The authors mainly summarized a POUNDS Lost trial, a 2-year study on overweight adults, which showed no significant difference in weight loss across various diets, and identified multiple intrinsic factors that may influence weight loss and cardiometabolic responses to dietary interventions. Several areas need further refinement and consideration by the authors: 

1. While the POUNDS Lost trial and other key studies discussed are foundational, they are over a decade old. For this review, the authors should focus on recent advancements in the field, rather than primarily featuring the authors' own prior research

2. The authors' overemphasis on the POUNDS Lost trial narrows the review's perspective. A comprehensive overview of the factors influencing weight loss and metabolism, drawn from a broader range of studies in the field, is necessary for the review

3. For Figure 1, the representation of the numbers on the X-axis is unclear, necessitating clear elaboration or revision

4. The current quality of the figures is not good and requires substantial enhancement to meet professional standards

5. In the discussion, the authors should explicitly delineate the potential future directions for the field and critically evaluate the limitations inherent in existing studies

Author Response

The authors mainly summarized a POUNDS Lost trial, a 2-year study on overweight adults, which showed no significant difference in weight loss across various diets, and identified multiple intrinsic factors that may influence weight loss and cardiometabolic responses to dietary interventions. Several areas need further refinement and consideration by the authors: 

We appreciate the reviewer’s comments, which are helpful in improving the quality of this review! Below please find our point-to-point responses. We have also revised the review article according to the comments.

  1. While the POUNDS Lost trial and other key studies discussed are foundational, they are over a decade old. For this review, the authors should focus on recent advancements in the field, rather than primarily featuring the authors' own prior research

We thank the reviewer’s comments. This is an invited review for specifically summarizing the major findings related to precision dietary interventions from the POUNDS Lost trial; therefore, many other studies which are not relevant to this purpose of the review are not included. In addition, this review is not to report the findings from the original trial; instead, it is to review the new findings with a focus on the results mostly published in the past few years.

  1. The authors’ overemphasis on the POUNDS Lost trial narrows the review’s perspective. A comprehensive overview of the factors influencing weight loss and metabolism, drawn from a broader range of studies in the field, is necessary for the review

We thank the reviewer’s comments. Please refer to our response to the first comment. We agree with the reviewer that to comprehensive overview of the factors influencing weight loss and metabolism, drawn from a broader range of studies in the field is important, but beyond the scope of this review.

  1. For Figure 1, the representation of the numbers on the X-axis is unclear, necessitating clear elaboration or revision

Thank you for the suggestion. We have modified the figure to make the representation of the numbers on the X-axis clearer.

  1. The current quality of the figures is not good and requires substantial enhancement to meet professional standards

We have modified the figures to improve the quality.

  1. In the discussion, the authors should explicitly delineate the potential future directions for the field and critically evaluate the limitations inherent in existing studies

We have modified the discussion accordingly.

Reviewer 2 Report

The authors present a review of the GWAS-related data from the POUNDS Lost trial. The review provides novel insights to the potential of precision weight management treatment for individuals as well as potential biomarkers for long-term weight loss success. 

Suggestions for improvement:

- use patient first language throughout, i.e. adults with obesity versus obese adults

- some of the data presentation is lacking transition and seems to jump around a bit. For example, describing PA data and jumping right into sleep. Please fix transitions throughout the manuscript

- line 372 "may not be"

- Figure 1: assuming light purple is baseline and dark purple is 6 months? Please label to make more clear to reader

- the inclusion of figure 3 is great to show analysis, would it be possible to include a figure that shows overall take home for current understanding of precision weight management based on review?

Author Response

The authors present a review of the GWAS-related data from the POUNDS Lost trial. The review provides novel insights to the potential of precision weight management treatment for individuals as well as potential biomarkers for long-term weight loss success

We appreciate the reviewer’s positive comments!

Suggestions for improvement:

- use patient first language throughout, i.e. adults with obesity versus obese adults

We have modified the text accordingly.

- some of the data presentation is lacking transition and seems to jump around a bit. For example, describing PA data and jumping right into sleep. Please fix transitions throughout the manuscript

We have modified the text to improve transitions as suggested.

- line 372 "may not be"

It was corrected.

- Figure 1: assuming light purple is baseline and dark purple is 6 months? Please label to make more clear to reader

We have added the label to Figure 1.

- the inclusion of figure 3 is great to show analysis, would it be possible to include a figure that shows overall take home for current understanding of precision weight management based on review?

As suggested, we added Figure 4 to show take home messages based on the review.

Round 2

Reviewer 1 Report

The authors has addressed all of my comments - thanks for the clarification. 

Reviewer 2 Report

Edits have satisfactorily met previous comments.